# EVALUATING ONLINE CONTINUAL LEARNING WITH CALM

## ABSTRACT

Online Continual Learning (OCL) studies learning over a continuous data stream without observing any single example more than once, a setting that is closer to the experience of humans and systems that must learn "on-the-wild". Yet, commonly available benchmarks are far from these real world conditions, because they explicitly signal different tasks, lack latent similarity structure or assume temporal independence between different examples. Here, we propose a new benchmark for OCL based on language modelling in which input alternates between different languages and domains without any explicit delimitation. Additionally, we propose new metrics to study catastrophic forgetting in this setting and evaluate multiple baseline models based on compositions of experts. Finally, we introduce a simple gating technique that learns the latent similarities between different inputs, improving the performance of a Products of Experts model.

## 1 INTRODUCTION

Machines, like humans, can learn to perform multiple different tasks from feedback alone (Caruana, 1997). On the other hand, humans, but not machines, can benefit from settings in which tasks are presented repeatedly for multiple trials before switching to the next one (Flesch et al., 2018), whereas machines require examples to be presented in a shuffled (*i.i.d*) order to learn effectively. Otherwise, they suffer from an effect known as "catastrophic forgetting" or "catastrophic interference" (McCloskey & Cohen, 1989; Ratcliff, 1990). While there has been a considerable amount of work focused on solving this problem, an endeavour that also goes by the name of 'Continual', 'Incremental' or 'Life-long' Learning, a large part of it is evaluated on settings in which there is an explicit delimitation signal for every new task presented to the model (Kirkpatrick et al., 2017; Zenke et al., 2017; Sodhani et al., 2018; Serra et al., 2018; Lopez-Paz & Ranzato, 2017; Fernando et al., 2017; Lee et al., 2017; Rusu et al., 2016; Li & Hoiem, 2018; Aljundi et al., 2017; Adel et al., 2020; Titsias et al., 2020; Ebrahimi et al., 2020; von Oswald et al., 2020; Li et al., 2020; Yoon et al., 2020). However, humans do not need any such signalling at all. Consider, for example, the case of a child growing up in a multi-lingual environment. Even though it is not entirely clear whether the child would rely on environmental cues (for instance, the identity of the speaker) to distinguish different input languages or not (De Houwer, 2017), any mechanism must be necessarily inferred from the context. Moreover, even the concept of "task" could be vacuous, as it could be represented by shifting data distributions (Lesort et al., 2020).

Even though the emerging field of Online Continual Learning (Parisi & Lomonaco, 2020; Aljundi et al., 2019a) or Task-Free Continual Learning (Aljundi et al., 2019b; Lee et al., 2020) has started to propose solutions to these problems, commonly available benchmarks make assumptions that are far from the real world conditions, such as lacking latent similarity structure on the data stream (e.g. orthogonal permutations of an image pixels) or assuming temporal independence between different examples (e.g. an image of a chair can be classified as "chair" independently of any previous examples). Consider, instead, the challenge of natural language learning which requires making sense of a highly correlated and temporally interdependent data stream. We argue that the notable scarcity of benchmarks featuring temporally correlated sequences of examples, with short and long-term dependencies, latent similarities between different classes of examples, and no explicit delimitation when transitioning between different classes has left a blind spot in the Online Continual Learning community, which we address here. Moreover, almost none of the commonly used benchmarks deals with language, further limiting the amount of research that extends to this modality.

Here, we make a two-fold contribution towards studying online continual learning in neural networks in a linguistic setting. First, we bring CALM (*Class-Agnostic Continual Language Modelling*) to the community, a continual language modelling evaluation framework containing text that alternates between different classes of input (e.g. different languages or domains) with latent similarities to which the models could adapt. We introduce two variants. The first is a character-based language modelling benchmark featuring five different languages that randomly switch between one another. The second one is a word-based language modelling task, where the text alternates between four different domains. No segmentation signal is given when a switch happens, thus requiring models to learn to adapt to these changes. We also propose novel metrics that capture the impact of catastrophic forgetting in an online learning setting by measuring how efficiently can models adapt to class switches. In line with Aljundi et al. (2019b), we note that when a distribution shift occurs, a neural network that suffers from catastrophic forgetting will display a spike in the loss signal, even when the distribution had been observed in the past (see Figure 1a). Thus, we propose catastrophic forgetting metrics based on characterizing the size of these peaks. The benchmark is provided as a Python library that can be easily imported into a PyTorch project.[1] Second, we evaluate multiple baselines based on expert architectures and propose a novel albeit simple mechanism that we call *plastic gates*, which we show to improve the performance of Products of Experts. Our post-hoc analysis shows that this mechanism is effective in producing a gating strategy that helps to circumvent catastrophic interference while also uncovering latent similarities in the input classes.

## 2 RELATED WORK

The field of Continual Learning, Incremental Learning or Lifelong Learning has grown to encompass a large body of work, which is better summarized in respective reviews (Parisi et al., 2019; Lesort et al., 2020). An overwhelming majority of this work concerns image classification problems or object recognition. Some evaluation datasets are derived from traditional machine learning datasets by manipulating the input examples in more or less artificial ways –like Permuted MNIST (Kirkpatrick et al., 2017) or Rotated MNIST (Lopez-Paz & Ranzato, 2017)– while others keep examples unchanged but present them in a specific non-i.i.d. order, like for instance, iCIFAR-100 (Rebuffi et al., 2017) or split-MNIST (Zenke et al., 2017). All of these datasets comprise single-input classification problems in which there are no temporal dependencies nor correlations between two consecutive examples. To better approximate the conditions of real-world experiences, Fanello et al. (2013), Lomonaco & Maltoni (2017) and Roady et al. (2020) introduced iCubWorld, CORe50, and Stream-51 respectively, which comprise short videos of objects from different angles (further including naturalistic scenes in the latter case). These datasets address the problem of correlated examples, but not of temporal dependencies, which we do address in this work. Li et al. (2020) and de Masson d'Autume et al. (2019) proposed the only benchmarks dealing with language that we know of, in which the former adopts a sequence to sequence paradigm to study incremental learning of new vocabulary items on simplified or artificial datasets, while the latter adapted existing text classification and QA benchmarks analogously to above-mentioned work in image classification. Our work instead uses naturalistic textual data containing natural latent similarities between distributions that can drive information transfer or forgetting.

By and large, work directed to address catastrophic forgetting in neural networks presumes the existence of a task identifier to signal different learning units. However, recent work has aimed at tackling catastrophic forgetting even in conditions in which no task boundaries are provided (Aljundi et al., 2019b; Lee et al., 2020), going under the name of "Task-Free Continual Learning" or "Online Continual Learning" (Parisi & Lomonaco, 2020; Aljundi et al., 2019a). Of these works, only Aljundi et al. (2019b) uses naturalistic data to classify actors appearing in soap-opera episodes (Aljundi et al., 2016), while others resort to artificially modified datasets like split or permuted MNIST. Here, we complement this resource with a text-based benchmark for Task-Free Continual Learning, while arguing for more work on more naturalistic non-i.i.d. datasets.

Another aspect of Continual Learning deals with how models are evaluated. Most often, this is done by measuring accuracy on a dedicated test set (Lopez-Paz & Ranzato, 2017; Díaz-Rodríguez et al., 2018; Hayes et al., 2018; Chaudhry et al., 2018; de Masson d'Autume et al., 2019). However this

---

[1]Code and materials included in the supplementary materials will be made publicly available upon acceptance.

evaluation protocol is tailored for batch learning conditions, in which a model is fit to a training dataset, and then stops learning. Here, instead, we argue in favour of situated evaluation protocols adapted to far-from-equilibrium learning conditions (Holland, 1992) by adopting an Online Learning framework (Hoi et al., 2018), which is also known as the prequential approach (Dawid, 1984; Gama et al., 2013).

On the modelling side, this work explores Mixture of Experts (Jacobs et al., 1991) and Product of Experts (Hinton, 1999) architectures. Variations thereof are at the base of many architectural proposals for addressing catastrophic forgetting (Rusu et al., 2016; Li & Hoiem, 2018; Aljundi et al., 2017; Lee et al., 2020). However, often they are accompanied by other mechanisms, such as the growth of new modules, freezing of weights or generative modelling of the input. Here we examine the simplest enactments of these architectures and propose an easy-to-implement gating mechanism which can be learned online and provides a strong baseline for more complex architectures.

Finally, our study falls within the line of language modelling using neural network models (Bengio et al., 2003; Mikolov et al., 2010). In this context, adaptation to the recent past has been studied in the context of cache models (Grave et al., 2017; Merity et al., 2017). There, a non-parametric model deals with capturing high-frequency recent statistics while a parametric model captures the more stable aspects of the distribution. These solutions, however, are not well-adapted for cases in which the whole distribution changes over time. Moreover, language modelling is generally studied in a train-test split, where a model is fitted to the training data and asked to generalize over the unseen test data. Here, instead, we study how a model can adapt to incoming data in an online fashion.

## 3 THE CALM BENCHMARK

We designed a benchmark for evaluating Online Continual Learning algorithms having in mind the following three desiderata: 1) naturally correlated sequential data, 2) task agnosticism and 3) temporally situated evaluation. Parisi & Lomonaco (2020) discusses the first two. The first requires that on the one hand, data is observed in a potentially infinite data stream with high-dimensional, non-stationary, and temporally correlated examples. The second, that learning systems should not be fed external task boundaries to help them learn in these conditions. Furthermore, we also introduce a third desideratum, by which we ask models to be evaluated *in-situ* on each example presented to the model, following the classical Online Learning setting (Hoi et al., 2018; Sahoo et al., 2018). We thus propose an Online Continual Learning benchmark featuring a language modelling objective where the data stream can switch between different distributions. Because switches are not announced to the model, this is a "Single-Incremental-Task" or "No task label" scenario under the framework proposed by Lesort et al. (2020).

More precisely, consider a sequence of observations $x_t \in \mathcal{X}$ that are fed to a model $M$ parametrized by $\Theta_t$, which makes the prediction $\hat{y}_t \in \mathcal{Y}$. Then, the true target $y_t \in \mathcal{Y}$ will be revealed and the loss $L_t = L(\hat{y}_t, y_t)$ is observed and later used to compute the model's performance from a given time $S$ until time $T$ as the average loss in that span $\bar{L}_S^T = \frac{1}{T-S} \sum_{t=S}^{T} L_t$ for evaluation purposes. Only after reporting the loss can the model be trained based on the received feedback, preventing data leakage. In practice, these examples are presented as mini-batches $(X_t, Y_t) \in \mathcal{X}^{b \times w} \times \mathcal{Y}^{b \times w}$ containing $b$ parallel streams, and chunked into small windows of length $w$ for efficiency considerations related to the training of neural networks (Parisi & Lomonaco, 2020).

The data stream is composed of $N$ sequences of consecutive mini-batches of length $T_1, T_2, \ldots, T_N$, and starting at positions $S_i = \sum_{j=1}^{i-1} T_j$. In turn, each of these sequences belong to one of $n$ different classes $[\mathcal{D}_1, \ldots, \mathcal{D}_n]$, presented in random order.

To characterize the effect of forgetting we note that a model that becomes disadapted to a given distribution will display a spike in the loss after the stream switches to this distribution, even if it has been observed before (see Figure 1a). For a model to be resilient to catastrophic forgetting, it must adapt quickly to every new distribution, which can be characterized by the height and width of these peaks. We thus propose the following metrics to complement the standard online performance:

- **Loss after switch**: Tracks the loss for the first $k$ times-steps after a switch occurs to quantify the height of the peak. Formally, L@sw $= \frac{1}{N} \sum_{i=1}^{N} \bar{L}_{S_i}^{S_i+k}$

- **Recovery time after switch**: Counts the number of time-steps that it takes the model to reach the mean loss observed for the last seen sequence of the current class. In this way, we can quantify the length of the peak.

## 3.1 DATASET

In this work, we created two datasets for CALM. One is character-level and multilingual, whereas the other is word-level and multi-domain. Both benchmarks feature conflicting learning signals when moving between domains or languages, making the learning systems susceptible to catastrophic forgetting.

For our first dataset (**MultiLingual** and character-based), we propose a language modelling benchmark in which incoming text data can alternate between different languages. This benchmark is character-based because there would hardly be any forgetting at the word level, as the word distributions hardly share any support. Concretely, we build on parts of the news corpus developed for the 2009 Workshop of Machine Translation (Callison-Burch et al., 2009). We extracted text from five languages: English, French, Spanish, German, and Czech (containing 1.8B, 572M, 160M, 715M and 439M characters, respectively) because they all have similar character sets, while also showing interesting linguistic variability. In particular, they belong to three different Indo-European branches: Romance (French and Spanish), Germanic (English and German), and Slavic (Czech). As a consequence, there is a latent similarity structure between the different classes that models could learn to recognize. Compared to earlier multilingual corpora (Kawakami et al., 2017), our dataset was carefully constructed to include only linguistically valid characters, in order to prevent non-linguistic noise from interfering with our experiments. For this, we removed all lines from the input that contained characters appearing less than 100 times on the full corpus. The resulting character vocabulary consists of 211 characters.

The second dataset is an English word-level **MultiDomain** dataset. For this, we used four different source corpora: news (same as above), europarl (Koehn, 2005), the British National Corpus (Consortium et al., 2007), and Wikipedia (Merity et al., 2017). They each have 300M, 54M, 100M and 101.4M tokens, respectively. In contrast with the previous dataset, word-level is the most appropriate choice here, as differences between the distributions at the character level would be too nuanced to drive any forgetting. We kept in the vocabulary the top 25K words for each corpus, which after merging yielded a vocabulary size of 58K words. Samples from all source corpora are included in the appendix.

We then created the final MultiLingual and MultiDomain corpora by joining $N = 100$ different fragments evenly distributed among the different classes (languages or domains) with lengths sampled from a (truncated) exponential distribution: $T_i \sim Exp(\lambda)$. Thanks to this distribution's memorylessness property, it is virtually impossible to estimate when the next switch is going to happen. While we do not constrain switches to occur at word or sentence boundaries, but rather after an integer number of sequences of length $w$, the noise introduced at transition points for this reason is relatively mild and does not affect the distribution-alternating nature of the dataset. In benefit, training and further analysis become considerably simplified by removing the need to handle variable-length input. We constructed two different variations with shorter or longer fragments. For MultiLingual, we constructed 1M and 10M-characters-long corpora with expected fragment lengths of $\lambda = 10k$ and $\lambda = 100k$ characters, respectively. For MultiDomain we followed the same procedure, extracting 100 alternating sequences with mean lengths of $\lambda = 10k$ and $\lambda = 20k$, for a total of 1M and 2M words. These relatively modest sizes allow for faster iteration and exploration of different models, while still allowing us to observe forgetting (or lack thereof) dynamics in the studied models. To facilitate further research, we release a Python library[2] providing a data iterator for both datasets in which a researcher can experiment with different variations by picking parameters $N$ and $\lambda$.

## 4 BASELINE MODELS

To endow CALM with simple and yet, strong baselines, we explored architectures based on (Weighted) Product of Experts or PoE (Hinton, 1999) and Mixture of Experts or MoE (Jacobs et al., 1991; Eigen et al., 2013), henceforth generically denoted expert architectures. Thanks to

---

[2] Available at `http://anonymized`

combining predictions from different experts, they can potentially learn different parts of the latent distributions. Moreover, gating weights can avert catastrophic forgetting on the individual experts by modulating the learning signal, making them an excellent candidate to model Online Continual Learning problems. Indeed, while variations thereof have been explored before (see Section 2), here we emphasize simplicity as it would befit baseline models, yet not neglecting performance.

In the standard implementation of expert architectures, mixture weights are produced by a third "gating" module as a function of the current inputs. While this gating model could quickly adapt to changes in the environment, learning to do so is far from trivial in a continual learning setup, sometimes requiring pre-training to distinguish input classes (Aljundi et al., 2017). The problem comes from the fact that the gating network must learn a latent classifier to pick the experts best adapted to the current class, but classes are observed non-i.i.d. as long sequences of examples from one class at a time. Thus, the gating network can easily settle for a constant function for any given current class, which only changes when examples of a different class start to be observed, making experts vulnerable to catastrophic forgetting. In order to alleviate this issue and make experts more stable, we propose *plastic gates*, by which the gates are fast-adapting parameter values that are trained on recent experience.

More formally, an expert architecture is composed of a set of modules $\mathcal{M} = \{M_1, \ldots, M_n\}$ with parameters $\Theta_{M_1}, \ldots, \Theta_{M_n}$, used to compute a unique prediction as follows. When an input $x$ (with target $y$) is observed, it is fed to all modules $M_{1 \ldots n}$, obtaining log-linear outputs $\tilde{\mathbf{y}}^{(1)} = M_1(x), \ldots, \tilde{\mathbf{y}}^{(n)} = M_n(x)$. Then, an additional vector of mixture weights $\mathbf{w} \in \mathbb{R}^n$ is used to combine them. This vector is computed by a separate gating module $\mathbf{w} = G(x)$ with parameters $\Theta_G$, jointly trained with the rest of the network. The output of the full model $\mathbf{y}$ is then a linear combination of the individual modules outputs $\tilde{\mathbf{Y}} = [\tilde{\mathbf{y}}^{(1)}, \ldots, \tilde{\mathbf{y}}^{(n)}]$ weighted by $\mathbf{w}^3$, after or before normalizing, depending on whether the model is MoE or PoE:

$$\tilde{\mathbf{y}}^{\text{MoE}}(\mathbf{w}) = \sum_{i=1}^{n} \text{softmax}(\mathbf{w})_i \left( \text{softmax} \, \tilde{\mathbf{y}}^{(i)} \right) \qquad \tilde{\mathbf{y}}^{\text{PoE}}(\mathbf{w}) = \text{softmax} \left( \tilde{\mathbf{Y}}^{\intercal} \mathbf{w} \right)$$

Note that in contrast to MoE, PoE are more efficient to compute because they do not require to normalize the output of each individual model. Once the loss is computed on a mini-batch $(X_t, Y_t)$ and kept for evaluation (see Section 3), all sub-networks $G$ and $\mathcal{M}$ are trained for one or more gradient steps to reduce this loss, and the system moves to the next mini-batch.

**Plastic Gates** Rather than learning a gating network, which can be challenging, we propose to continually learn the gating coefficients that best fit the recent experience:

$$\mathbf{w_{t+1}} = \arg\min_{\mathbf{w}} L(\tilde{Y}_t(\mathbf{w}), Y_t)$$

In practice, we perform a (hyperparameter) number $k$ of gradient descent steps on the above objective to allow for some regularization of the gates over time.

**Parametrization for Language Modelling** We instantiate the expert modules $M_i$ to be double-layered LSTM networks (Hochreiter & Schmidhuber, 1997), with predictions $\tilde{\mathbf{y}}_{\mathbf{t}}^{(i)}, \mathbf{h_{t+1}}^{(i)} = \text{LSTM}_i(x_t, \mathbf{h_t}^{(i)})$. For the regular gating network, we use a single-layer LSTM network. That is, $\mathbf{w_t}, \mathbf{h'_{t+1}} = \text{LSTM}(x_t, \mathbf{h'_t})$.

## 5 EXPERIMENTS

We explored the performance of different baseline models while they made a single pass over the CALM datasets. Following standard practice, rather than reporting the cross-entropy loss, we use the perplexity at each time step, given by $exp(L_t)$. Furthermore, we allowed the models to learn over the first half of the datasets without being evaluated, and only start computing metrics on the second half. Otherwise, we use the measures discussed in Section 3 to track models' performance, namely, average perplexity (**ppl**), average perplexity for $k = 10$ batches after a switch (**ppl@sw**) and recovery time after a switch (**rec**).

---

[3]Note that the since $\tilde{y}^{\text{PoE}}$ linearly combines the logits is is effectively computing a geometric combination of each individual module's unnormalized probabilities: $\exp(\tilde{\mathbf{y}}_j^{\text{PoE}}) \propto \prod_{i=1}^{n} \exp(\tilde{\mathbf{y}}_i^{(j)})^{w_i}$.

We explored models featuring different degrees of modularization, varying their hidden size vectors to make them all have an approximately equal total number of parameters. On one extreme, we had a large two-layers **LSTM** network. Next, we considered standard **PoE** and **MoE** models with mixture weights computed by an LSTM gating network, plus their plastic weights variants (**+PW**), as described in Section 4. Moreover, we trained ensemble models (**Ensemble**), which are equivalent to a MoE where all mixture weights are $\frac{1}{n}$ for all $n$ modules. We studied both a more centralized network composed of 5 modules and larger hidden dimensionality (marked with **5**) and a more distributed network with 30 modules but with smaller hidden sizes (marked with **30**). As reference points (but not as real contenders), we also trained independent LSTMs (**Ind. LSTM**), one for each class, which enabled us to compare the performance of our model to a situation where there is no forgetting from conflicting learning signals, but also where there is no possibility of transferring learned representations across possibly related domains. Furthermore, we compare a Mixture-of-Softmax (**MoS**) model (Yang et al., 2018), in which multiple softmax layers are combined to extract the predictions from a single LSTM module. While we were also interested in applying state-of-the-art online continual learning methods (Lee et al., 2020; Aljundi et al., 2019a), having these systems being designed for image classification datasets they would require non-trivial adaptations significantly departing from the original models, which would limit any possible conclusions we could draw. Similarly, we experimented extensively on validation data with Transformer models (Vaswani et al., 2017). However, due to these models sensitivity to dataset size and learning rates scheduling schemes which have been studied extensively for batch-learning (Popel & Bojar, 2018), but not for these far-from-equilibrium (Holland, 1992) conditions, their performance was worse than expected. We give a detailed account of our attempts in the appendix and leave a study on how to adapt these models for Online Continual Learning for future work.

We controlled the number of model parameters to remain constant for each of the MultiLingual (about 21M parameters) and the MultiDomain (about 600M parameters) experimental setups. (The difference in size is explained by the larger vocabulary sizes in the latter.) For this, we adjusted the hidden dimensionality of different models accordingly, which, together with all explored hyperparameters, are reported in the appendix. We kept the size of the incoming batches fixed at $w = 20$ and $b = 10$ for all models and used PyTorch (Paszke et al., 2017) with the standard implementations for the underlying models.

## 5.1 RESULTS

| | MultiLingual | | | | | | MultiDomain | | | | | |
|---|---|---|---|---|---|---|---|---|---|---|---|---|
| | $\lambda = 10$k | | | $\lambda = 100$k | | | $\lambda = 10$k | | | $\lambda = 20$k | | |
| | ppl | ppl@sw | rec | ppl | ppl@sw | rec | ppl | ppl@sw | rec | ppl | ppl@sw | rec |
| Ind. LSTM | 7.1 | 7.16 | 1.15 | 4.7 | 4.73 | 1.18 | 356 | 349 | 1.11 | 295 | 292 | 1.15 |
| Large LSTM | 7.78 | 10.4 | 6.82 | **4.86** | 8.58 | 18.9 | 352 | 406 | 3.61 | 457 | 619 | 6.56 |
| MoS | 8.13 | 10.6 | 6.6 | 5.43 | 10.3 | 19 | 343 | 443 | 4.6 | 298 | 409 | 6.08 |
| Ensemble 5 | 8.84 | 11.3 | 7.41 | 5.6 | 10.2 | 24.7 | 418 | 519 | 3.89 | 317 | 411 | 4.83 |
| MoE 5 | 8.65 | 10.9 | 7.11 | 5.55 | 9.86 | 24 | 425 | 524 | 3.76 | 335 | 439 | 4.94 |
| MoE+PW 5 | 8.74 | 11.1 | 7.2 | 5.58 | 10 | 23.3 | 446 | 557 | 3.94 | 331 | 432 | 4.63 |
| PoE 5 | 7.68 | 10.1 | 7.06 | 5.32 | 9.79 | 25.5 | 297 | 389 | 5.18 | 404 | 505 | 4.47 |
| PoE+PW 5 | **7.2** | **8.46** | **3.67** | 5.02 | 7.54 | 14.9 | 320 | 361 | 2.82 | 270 | 322 | **3.35** |
| Ensemble 30 | 11.9 | 14.8 | 8.08 | 7.05 | 14.2 | 30.9 | 509 | 623 | 3.72 | 391 | 511 | 5.14 |
| MoE 30 | 11.1 | 13.7 | 7.54 | 6.89 | 13.7 | 30 | 539 | 651 | 3.47 | 436 | 572 | 4.97 |
| MoE+PW 30 | 11.2 | 13.8 | 7.97 | 6.92 | 13.7 | 29.7 | 555 | 675 | 3.49 | 419 | 561 | 5.43 |
| PoE 30 | 7.96 | 10.7 | 7.33 | 5.17 | 9.9 | 24.8 | 315 | 375 | 3.89 | 297 | 389 | 5.18 |
| PoE+PW 30 | 7.41 | 9.17 | 4.76 | 5.04 | **7** | **9.03** | **285** | **316** | **2.68** | **241** | **287** | 3.54 |

Table 1: Average perplexity (ppl), perplexity for 10 batches after a switch (ppl@sw), and recovery time after a switch in batches (rec) for both datasets per mean sequence length ($\lambda$).

Results are averaged over ten different runs and reported in Table 1. Standard deviations are reported on the Supplementary Materials.

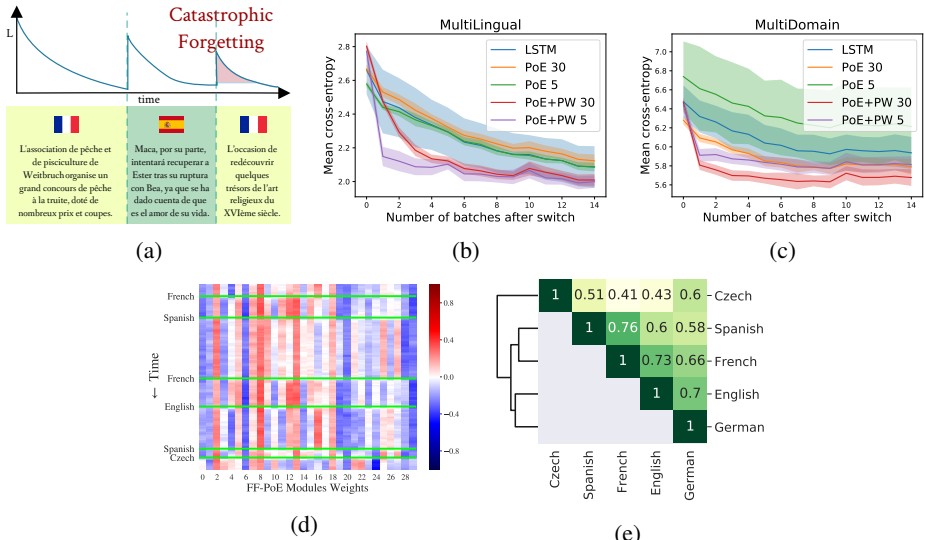

Figure 1: (a) CALM: A model's loss ($L$) is tracked as it observes text pertaining to different classes, while catastrophic interference provokes spikes in this signal. (b-c) Mean cross-entropy for the first 15 batches after a switch averaged over all occurrences in MultiLingual and MultiDomain, respectively, under different random seeds. (d) Mixture weights produced by the PoE+PW 30 model on multilingual data ($\lambda = 10$k). (e) Correlation coefficients between mixture weights corresponding to different languages for the PoE+PW 30 model collected during the last 100 batches ($\lambda = 10$k).

We begin by observing that higher values of $\lambda$ correspond to lower perplexities, as expected from the fact that these corpora with longer sequence lengths are also proportionally larger in total length.

Second, we note that Ensemble and MoE systems with 5 modules and larger hidden vectors outperformed models with 30 modules and smaller hidden dimensionality, but this is not the case for PoE, which show comparable performance between the two variants or even the opposite trend. Furthermore, the PoE's performance is considerably better than the former two, which can be attributed to a combination of multiple factors. On the one hand, we note that, in agreement with previous work (Shen et al., 2019), MoE models are often sensitive to a Winner-Takes-All (WTA) effect in which only one single expert gets trained at the end. Thus, models with larger dimensionality per module can benefit from having a larger capacity. However, also ensembles show comparable performance, showing that this effect is not only caused by a single module being trained. Perhaps, more important is the fact that, as hypothesized by Hinton (1999), PoE can use their capacity to learn complementary parts of the distribution, and thus it makes a smaller difference for them whether there are a few high-capacity modules or many of them, but with smaller capacity.

Next, we note that while PW does not strongly alter performance on MoE architectures, as expected from the WTA effect influencing these models, they significantly improve the vanilla PoE counterparts, confirming the effectiveness of the proposed mechanism in this task. This observation holds not only for overall perplexity but also in terms of the metrics quantifying adaptation efficiency at class switches (ppl@sw and rec). Indeed, Figures 1b and 1c show this fact in more detail, by representing the mean cross-entropy of each different model for the 15 batches occurring immediately after a switch. As we can see, the PoE+PW model shows a large spike on the first batch because its adaptation mechanism that depends on this error signal has not kicked in yet. However, in the subsequent batch, its performance increases sharply outperforming comparable models.

In comparison to a monolithic LSTM model, PoE and PoE+PW models perform on-par on Multi-Lingual (although with better adaptation records), and better on MultiDomain. In the latter case, we can observe that the version with 30 modules yields better performance than the one with just 5. One possible explanation is related to word-level language modelling being a higher rank problem than character-level language modelling, and thus it can be better fitted by combining the judgements from multiple lower-rank experts (Yang et al., 2018). This explanation is also consistent with the comparatively better performance of the MoS model.

Finally, we note that the model with Independent LSTMs for each class performs best on MultiLingual, but it is outperformed by a large margin on MultiDomain. We note that this model does not suffer from forgetting when switching classes but also misses the training signal from transferable training data. As a consequence, it has an edge on MultiLingual, which switches between classes that have considerably different statistical properties, but not on MultiDomain where the differences between the classes are much more nuanced. All in all, this shows that while, in consonance with previous results (Dhar & Bisazza, 2018), there is little room for transferring knowledge in the MultiLingual case, the MultiDomain setting provides plenty of opportunities for transferring knowledge across each domain. Thus domain-agnostic systems can benefit from them.

## 5.2 Analysis

Next, we turned to analyze the gating strategies acquired by the more successful models to understand whether they have captured the latent similarities between classes and how might they help them in coping with catastrophic forgetting. For this, we focused on the PoE+PW 30 modules operating on the MultiLingual dataset ($\lambda = 10k$) because its 30-dimensional gate vectors can represent more nuanced similarities.

Figure 1d shows a heatmap of the mixture weights as the model processes different language sequences. High absolute values represent the activation of a module, regardless of whether these are negative or positive values. It can be seen that upon language switches, the model reconfigures itself to a different set of mixture weights that are maintained more or less consistently within the sequence. Furthermore, we note that modules that receive mixture weights close to $0$ are protected from forgetting, as this gating value is also multiplied to the module's gradients. Moreover, we hypothesize that modules are protected even when their corresponding weight is set to the opposite sign (see, for instance, module 16 on English and Spanish), because the incoming training data serves as negative training data, namely, something not-to-be-predicted. Thus, this should not affect what the model does predict when used with a positive weight. Instead, this allows for dual use of the modules, encoding information both when it is weighted positively and negatively.

Finally, recall from Section 3.1 that the languages in our MultiLingual dataset are derived from different linguistic families with a latent similarity structure. To uncover whether the learned latent similarities reflect this structure, we computed the correlations between the mixture weights produced while processing the last 100 batches of each class. The results are displayed in Figure 1e, and show that the similarities are indeed well-reflected in the gating values. Notably, we observe that Czech seems to be using the most distinct set of modules. Spanish and French correlate quite strongly in the modules they use, and while English also correlates with French, it also does so with German, with the latter correlating to a lesser extent with the other languages. Indeed, applying a simple hierarchical clustering algorithm over this matrix recovers the underlying linguistic families!

## 6 Conclusions

In this paper, we have introduced the class-agnostic continual language modelling task (CALM), together with a Python library with MultiLingual and MultiDomain datasets, which allows multiple parameter configurations and can also be easily adapted to different corpora. We expect that it will foster more empirical work on continual learning in a language-centred setup in which there is a natural latent similarity structure between different tasks. We have argued that in addition to measuring the overall performance of models in an online learning fashion, their susceptibility to catastrophic forgetting can be observed in terms of adaptation speed to changes in the input class, and proposed measures to capture it. Finally, we have evaluated multiple simple baselines to serve as references for future work on this benchmark and introduced a simplification of the gating strategy for a Product of Experts, which improves its performance significantly by allowing it to distribute effectively different distributions across different experts so that the resulting system can act as a strong baseline for future work on this task.

While addressing catastrophic forgetting is still a major challenge for Online Continual Learning, it is by no means the only one. In the future, we would like to understand how learning systems can also bootstrap on their knowledge to improve their learning skills, so that they will not only be able to acquire knowledge from different sources in a seamless way but also get better at it as they go.

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

## A    CORPUS EXAMPLES

Figure 2 and 3 present samples from the corpora used for our dataset. As stated in the paper, we can notice a much bigger difference between input class in the case of the multilingual setup, while the differences in the case of the multidomain setup are more subtle and nuanced.

|  | Dataset samples |
| --- | --- |
| czech | Maďarská iNFiNITY Coliseum Lan je pokračováním úspěšného BECUPu, z něhož si nejeden náš tým v minulosti odvezl medaili. |
| english | If Hofmann played the role of paterfamilias, Anaïs Nin was the bad mother to Admiral and De Niro's group. This one wasn't close. |
| french | Le Beatle s'en est alors emparé pour créer un chef-d'oeuvre psychédélique longtemps associé à l'usage du LSD. |
| german | Im ersten Jahr hatten sie schon 278 Anfragen, fast 60 ehemalige Manager und Unternehmer wollten mitmachen. |
| spanish | Los despidos serán realizados por medio del plan de GM de cese de empleo, por lo que no se ofrecerán jubilaciones anticipadas |

Figure 2: Samples from the multilingual dataset

|  | Multidomain dataset samples |
| --- | --- |
| bnc | Good weather for the crops. Have your sheep been suffering much from the staggers ? Have you contributed a great deal this year to the butter mountain ? |
| euro | I would like your advice about Rule 143 concerning inadmissibility. My question relates to something that will come up on Thursday |
| news | If Hofmann played the role of paterfamilias, Anaïs Nin was the bad mother to Admiral and De Niro's group. This one wasn't close. |
| wiki | Otto , Prince of Bavaria , was chosen as the first King of Greece in 1832 , under the name Othon . His arrival in Nafplio , then the Greek capital, was hailed enthusiastically by Makriyannis |

Figure 3: Samples from the multi-domain dataset

## B    FURTHER ANALYSIS

### B.1    PoE WEIGHTS BEHAVIOUR

We also inspected the gate values produced by LSTM-gated PoE models observing that the models are indeed not learning a class-switching mechanism. We hypothesized that this is due to the fact that when the experts are still untrained, the LSTM produces some arbitrary but consistent gating values, making those selected modules being the only ones to be trained, and thus falling into a vicious cycle. As a sanity check that supports this hypothesis, we first pre-trained a set of modules while still using our simple gating mechanism. Then, we initialized with these modules a network that now used LSTM mixture weights, but training on very short sequences to avoid the effect of catastrophic forgetting affecting the network. In this context, the network learned the appropriate gating as expected.

### B.2    MULTIDOMAIN MODULE CORRELATION

In comparison with the Multilingual setup, correlations in the MultiDomain case are much weaker. Moreover, they are weak even within the same class: When we measure the autocorrelation between

weights pertaining to the last 100 batches with the preceding 100 ones we obtain values in the order of 0.65, much lower than for the MultiLingual experiments, where they are in the order of 0.96 (see Figure 4b). This shows that model usage is less consistent per-class, which could be explained by the fact that classes are much more nuanced than in MultiLingual and their corresponding distributions are far more complex. These results are also consistent with our experimental observation that the MultiDomain dataset was more amenable to transfer between different classes than the MultiLingual, as these classes could be distributed more evenly across the model and could be characterized with multiple mixture weights configurations.

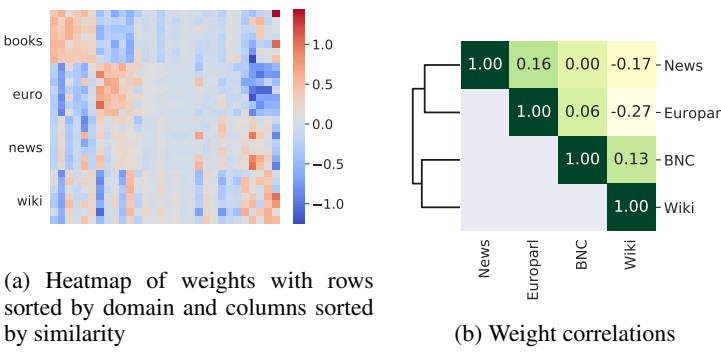

(a) Heatmap of weights with rows sorted by domain and columns sorted by similarity

(b) Weight correlations

Figure 4: MultiDomain ($\lambda$=10k) analysis

## C  TRANSFORMER EXPERIMENTS

We experimented extensively with Transformer models. One difference with respect to LSTM models is that Trasnformers, at least in their vanilla versions, are not autoregressive, and thus they cannot transfer information from the past. In standard NLP tasks, they largely overcome this problem by using a large context window on which they can operate effectively. Thus, to afford them similar memory capabilities, we kept a buffer of the last $b \times 512$ consecutive examples that was continually updated with each incoming mini-batch.

Vaswani et al. (2017) describes a learning rate scheduling scheme in which the learning rate is linearly increased until reaching a number of warmup steps, and then annealed from that point on. Considering that in a Continual Learning setup the model is not expected to converge, annealing might not be appropriate. Thus, we also experimented with keeping the learning rate flat after that point on. We experimented with both learning rate schedules, plus no scheduling at all. Furthermore, we considered both training with default Adam parameters ($\alpha = 10^{-3}, \beta = (0.9, 0.999)$) or the ones reported by Vaswani et al. (2017) and base learning rates of $\frac{1}{d_{model}}, 1e^{-3}, 5e^{-3}, 0.01$. Also, we tuned the warmup steps among 400, 2000 and 4000 steps. The best perplexity results we obtained in the Multilingual validation data were 13.2 for $\lambda = 10$k and 6.02 for $\lambda = 100$k, whereas in the Multidomain validation data we got 686 average perplexity for $\lambda = 10$k and 527 for $\lambda = 20$k.

## D  GENERATED OUTPUT

In Figure 5, we present generated samples from different stages of training. These generated examples are produced by sampling one character at a time from the models, and using them as input for the next time step. As quantitatively observed in the paper, it adapts much faster to the current input type (French) in comparison with an LSTM, which generates text resembling the language of the previously seen class even after 10 batches.

| | PoE+PW 30 | LSTM |
|---|---|---|
| end english | for a release was the week in Troust Pglates in George Services are claimed whet this could get one | Tvice. (Relátórs had the state's annual annual Call Statua plannting more years' physical cost |
| 5 batches french | lement Filmarian Roads. Aus cadres et temps disputer Lileana Maan. Institution, le provinces, unbieut | eau polítical but the room of Noxe Common Electrical Taladei Baritef. BAG - Runey premium begai maki |
| 10 batches french | Definit werde à l'équipe pass, libertant Youth Losier Chavez and Jean-Pierro. «Vu entre fascal publ | attempted Jueves Mo., unit encome ergarded a next post television genetical dangere tet. For hemous |
| end french | ive commune services au championnat où qui se sont renfovées de la hierre du 23,4er est dit doubles | el-Bilanze extranger à la fin de l'Etat: "Yens ni irneu à Show Joban ? Il vio, les grandes hommes de |

Figure 5: Generated text at different stages
of training

# E  MODEL SIZES

As it is shown in Table 2, the number of hidden units varies for most of the models. We vary the hidden size in order to keep a similar number of parameters across the models: around 22 million for the multilingual setup and around 600 million for the multidomain one.

| | MultiLingual | | MultiDomain | |
|---|---|---|---|---|
| Model | Hidden size | #Parameters | Hidden size | #Parameters |
| LSTM | 1300 | 21.66M | 5200 | 605.2M |
| Ind. LSTM | 550 | 20.2M | 1800 | 571.2M |
| PoE/MoE (+PW) 5 | 550 | 21.2M | 1600 | 621.8M |
| PoE/MoE (+PW) 30 | 200 | 21.85M | 200 | 635.3M |
| MoS | 500 | 22M | 2620 | 572M |

Table 2: Model sizes

# F  HYPERPARAMETER SEARCH

Table 3 present the explored hyperparameters for LSTM and PoE. The parameters in bold are the ones chosen for the final models, with the exception of PoE-5 and PoE+PW 5 which are marked with italics.

The meaning of the different hyperparameters for Table 3 is:

- nhid: the size of the hidden state of the base LSTM
- dropout: the dropout value used in the base module of the LSTM
- learn iter.: how many learning iterations over each batch are done before moving to the next batch
- adapt. iter.: it is used in the case of PoE+PW and it shows how many iterations to train the gating weights are done for each learning iteration.
- modules: how many modules does the PoE models contain
- gating nhid: the size of the hidden state for the LSTM used to calculate the gating weights in the case of PoE
- clear gating: it is a boolean value which clears the hidden state of the LSTM used for gating weights in the case of PoE

Also, MoS was tuned following the hyperparameters: 1 or 2 learning iterations, learning rate 1e-3 or 5e-4. For the domain setup, we considered the combinations: (nsoftmaxes=2, nhid=4750) or (nsoftmaxes=50, nhid=2620). On the other hand, for the multilingual dataset, we considered (nsoftmaxes=2, nhid=1200) or (nsoftmaxes=75, nhid=500).

| task | $\lambda$ | model | nhid | dropout | learn iter. | adapt. iter. | modules | gating nhid | clear gating |
|---|---|---|---|---|---|---|---|---|---|
| lang. | 10k | lstm | 200, **1300** | 0.1, **0.2**, 0.4 | 1, **2**, 5 | - | - | - | - |
| | | MoE/PoE | **200**, *550* | *0.2* | **2**, 5 | *1* | *5*, **30** | 50, *100* **200** | **0**, *1* |
| | | MoE/PoE+PW | 200, *550* | *0.2* | *2*, 5 | 1, **10**, *100* | *5*, **30** | - | - |
| | 100k | lstm | 200, **1300** | **0.1**, 0.2, 0.4 | **1**, 2, 5 | - | - | - | - |
| | | MoE/PoE | **200**, *550* | *0.2* | *1*, 2, 5 | *1* | *5*, **30** | 50, 100 *200* | 0, **1** |
| | | MoE/PoE+PW | **200**, *550* | *0.2* | *1*, 2, 5 | 1, 10, ***100*** | *5*, **30** | - | - |
| dom. | 10k | lstm | **5200** | **0.1**, 0.2, 0.4 | 1, 2, **5** | - | - | - | - |
| | | MoE/PoE | 200, *1600* | *0.2* | 1, **2**, 5 | *1* | *5*, **30** | *50*, 100 **200** | *0*, 1 |
| | | MoE/PoE+PW | **200**, *1600* | *0.2* | *1*, **2**, 5 | 1, 10, ***100*** | *5*, **30** | - | - |
| | 20k | lstm | 200, 1300, **5200** | 0.1, 0.2, **0.4** | **1**, 2, 5 | - | - | - | - |
| | | MoE/PoE | **200**, *1600* | *0.2* | *1*, 2, 5 | *1* | *5*, **30** | 50, 100 ***200*** | **0**,1 |
| | | MoE/PoE+PW | **200**, *1600* | *0.2* | **1**, 2, 5 | 1, 10, ***100*** | *5*, **30** | - | - |

Table 3: Table with the hyperparameters tested on the models: LSTM, PoE, and PoE+PW. The bold parameters are the ones chosen for LSTM, MoE/PoE-30, MoE/PoE+PW 30 and the italic parameters are the ones chosen for MoE/PoE-5 and MoE/PoE+PW 5

# G    STANDARD DEVIATIONS

| | MultiLingual | | | | | | MultiDomain | | | | | |
|---|---|---|---|---|---|---|---|---|---|---|---|---|
| | $\lambda = 10k$ | | | $\lambda = 100k$ | | | $\lambda = 10k$ | | | $\lambda = 20k$ | | |
| | ppl | ppl@sw | rec | ppl | ppl@sw | rec | ppl | ppl@sw | rec | ppl | ppl@sw | rec |
| Ind. LSTM | 0.42 | 0.41 | 0.44 | 0.12 | 0.05 | 0.3 | 28.5 | 25.5 | 0.2 | 17.1 | 16.6 | 0.22 |
| Large LSTM | 1.08 | 1.81 | 0.98 | 0.28 | 0.87 | 2.84 | 51 | 98.1 | 0.64 | — | — | — |
| MoS | 0.5 | 0.8 | 0.87 | 0.15 | 0.55 | 1.82 | 32.4 | 40.7 | 0.2 | 19.9 | 18.5 | 0.4 |
| PoE 5 | 0.23 | 0.2 | 0.7 | 0.14 | 0.24 | 2.51 | 229 | 332 | 0.32 | 18.8 | 19.6 | 0.5 |
| PoE 30 | 0.28 | 0.28 | 0.72 | 0.12 | 0.21 | 1.5 | 27.7 | 14.4 | 0.22 | 15.4 | 15.1 | 0.48 |
| PoE+PW 5 | 0.17 | 0.33 | 0.9 | 0.11 | 0.48 | 3.2 | 26.4 | 22.1 | 0.52 | 15.7 | 19.6 | 0.62 |
| PoE+PW 30 | 0.21 | 0.2 | 0.44 | 0.1 | 0.1 | 0.66 | 23.7 | 16.5 | 0.3 | 14.3 | 12.2 | 0.28 |
| Ensemble 5 | 0.249 | 0.301 | 0.85 | 0.12 | 0.35 | 2.32 | 54.5 | 76.4 | 0.419 | 27.6 | 32.6 | 0.407 |
| Ensemble 30 | 0.375 | 0.525 | 0.923 | 0.203 | 0.549 | 2.01 | 35.1 | 45.9 | 0.287 | 24.5 | 27.8 | 0.545 |
| MoE 5 | 0.255 | 0.274 | 0.855 | 0.12 | 0.426 | 1.75 | 64.2 | 84.4 | 0.447 | 42.4 | 56 | 0.464 |
| MoE 30 | 0.21 | 0.22 | 0.3 | 0.1 | 0.16 | 2.17 | 21 | 23.2 | 0.53 | 18 | 17.2 | 0.45 |
| MoE+PW 5 | 0.264 | 0.377 | 0.806 | 0.101 | 0.322 | 2.36 | 53.2 | 71.6 | 0.36 | 34.4 | 40.4 | 0.445 |
| MoE+PW 30 | 0.326 | 0.561 | 0.753 | 0.195 | 0.543 | 2.24 | 43.5 | 43.8 | 0.315 | 26.6 | 40.3 | 0.74 |

Table 4: Standard deviation for Average perplexity (ppl), perplexity for 10 batches after a switch (ppl@sw), and recovery time after a switch in batches (rec) for both datasets per mean sequence length ($\lambda$).

