# OpenReview forum: "Evaluating Online Continual Learning with CALM"
_ICLR.cc/2021/Conference — Reject_

### Official Review · AnonReviewer1 · 2020-10-26
**Useful dataset for online continual learning, evaluation could be improved with a wider range of baselines**

**Rating:** 6
**Confidence:** 4

**Review:**

This paper’s main contributions are (i) to propose two new benchmarks for online continual learning in the context of language modelling and (ii) evaluate the performance of a number of composition-of-experts-based models on the new datasets using a number of metrics. The multilingual benchmark, derived from an existing multilingual news corpus, consists of sequences of characters where the language is periodically switched, and the MultiDomain benchmark consists of sequences of English words where the corpus is periodically switched. The comparative performances of the various baselines on the two datasets, as well as an analysis of the mixture weights in one of the models during training, are used to provide insights into the qualitative differences between the datasets.

Overall, I am inclined to recommend acceptance for this paper on the margin because it makes a good contribution towards evaluating continual learning models in more real world settings, more specifically in the context of online learning. The datasets proposed are well-suited for purpose for reasons outlined below, and the evaluation using various composition-of-experts models is fairly conducted and followed up with an informative analysis. The key downside of the paper is that no standard continual learning baselines are trained on the proposed datasets; I would be inclined to increase my score if results were shown for 1 or 2 algorithms specifically designed for continual learning with neural networks (as discussed in more detail below).

Positives:
	•	There is a need to start evaluating continual learning in closer-to-real-life settings; in providing datasets that facilitate evaluation of continual learning models in an online setting without task boundaries, this paper makes a positive contribution in this direction.
	•	The datasets are simply composed, but seem well suited for evaluating online continual learning because (i) language data is sequential, (ii) by imposing a truncated exponential (and thus memoryless) distribution on the length of subsequences, it is hard for models to cheat in predicting the next task switch, preserving task-agnosticity, and (iii) in both datasets, the subtasks share latent similarities, creating the possibility for forward/backward transfer between them.
	•	The analysis of the experiments provides interesting insights into the datasets and differences between the baselines. E.g. Figure 1d effectively shows how the weights of one of the Product of Experts models switch after a task change, indicating a degree of specialisation of the modules, and 1e uses the correlations of the mixtures weights used for different subtasks to highlight the latent similarity between pairs of subtasks.
	•	The paper is clearly written and easy to follow.

Main Concern
	•	Limited set of baselines. While a range of composition-of-experts baselines are used for evaluation, it would have been much better to also include other methods specifically designed for online continual learning, such as those cited in the paper [1, 2] or, though not strictly online, a replay-based method such as CLEAR, which works in the task-agnostic setting. It is claimed in the paper that including state-of-the-art online continual learning methods would have involved “non-trivial adaptations significantly departing from the original models, which would limit any possible conclusions we could draw” as they are designed for image-based datasets. I don’t fully understand the basis of this claim; perhaps the authors could elaborate - as far as I am aware, for example, [1] is not restricted for use on image-based datasets.
	•	Since the subtasks do have discrete boundaries, even though these are not passed to the model during training, it would be possible to evaluate methods that use task boundaries for consolidation on the proposed datasets by either providing knowledge of the boundaries (although this breaks the task-agnosticity) or by using methods that can detect task boundaries - e.g. EWC uses the Forget-Me-Not Process [3].
	•	Overall, not evaluating the datasets with any standard continual learning baselines is an important weakness.

Other comments
	•	The proposed method, plastic gates, which performs best amongst the baselines used when combined with product of experts models, seems simple and effective but I am inclined to question how novel it is, since it just amounts to multi-step online gradient descent on the mixture weights.
	•	The metrics used for evaluating continual learning, loss after switch and recovery time after switch, which are one of the main selling points of the paper are suitable for the datasets provided, but would not be applicable in a setting where either the task boundaries are not known or there are no hard task boundaries to be identified.
	•	Typo Section 2 Paragraph 2: “MNNIST” -> “MNIST”

---

> ### Author Response · Authors · 2020-11-24
> **authors' response**
>
> We thank the reviewer for the detailed review. We appreciate the fact that the reviewer recognizes the qualities of the presented datasets, and of the complementary analysis.
> While we agree that it would be nice to have comparative results to other online continual learning algorithms, one of the main problems is that as current methods are designed for learning with single independent data instances, like in the case of labelled images. (We now highlight more explicitly this common assumption in the introduction.) Therefore, they assume, on the one hand, that a handful of examples can well characterize the data region that is not to be forgotten. On the other hand, every example is assumed to be independent of its previous context. Both of these assumptions are violated in the online continual language modelling benchmark that we are proposing. Furthemore, EWC assumes that after training on one task, we have found an optimal set of weights that we do not want to forget. In our case, however, the model will be exposed multiple times to the same underlying language/domain. Therefore, it would be completely unsurprising if EWC is not a good method for this setting. Plus, on top of requiring explicit labels it also needs a second pass over each example to compute the Fisher information, which is incompatible with the online setting. We can further think of other ways of trying to adapt existing methods to this setting, but as we argue these adaptations are not at all trivial, and represent research questions on their own right. Here, instead we focus on introducing a novel way of evaluating Online Continual Learning methods, which we hope would spur research on a more naturalistic setting that has been so far completely neglected.

---

### Official Review · AnonReviewer2 · 2020-10-28
**Official Blind Review R2**

**Rating:** 4
**Confidence:** 4

**Review:**

#############################################################################################

Summary:

This paper introduces a dataset and benchmark for language modeling in the online continual learning framework. The key characteristics of this benchmark are: the data are temporally correlated, there are no task identifiers presented to the model (task-free setting), and the evaluation is performed in an online fashion.

The benchmark consists of a Multilingual character-level dataset and a Multidomain word-level dataset. The authors introduce several metrics and evaluate using several simple baselines of mixtures-of-experts and products-of-experts.

#############################################################################################

Pros:

1.	The paper is clear and well-written.
2.	The authors provide sufficient details on data collection and modeling
3.	The relevant work section is extensive
4.	The design choices in constructing the dataset are well thought out and make sense given the objective of the paper. In particular, the dataset along with the proposed evaluation metrics captures the three stated objectives of the benchmark.
5.	The authors are upfront about materials left out of the main text. It’s nice when potential questions are anticipated and answered, for example, “why weren’t continual learning SOTA models evaluated?” and “why weren’t transformers considered as baselines?” The authors answer these questions candidly.

#############################################################################################

Cons

1.	The dataset seems incremental over existing work
2.	The introduced evaluation metrics are described intuitively, but are not analyzed empirically or theoretically
3.	The necessity/value of the introduced dataset is not adequately justified in relation to existing challenges in the continual learning setting. A component of this is showing where existing models fail (and why this dataset will help improve them).

#############################################################################################

Recommendation and explanation

I recommend rejection for the previously outlined reasons.

#############################################################################################

I also have some questions that I hope the author can help address:

1. What is the key innovation over existing work such as d’Autume et al. who also study language models in the continual learning, task-free setting?
2. What failure of current models does this benchmark address? Note that the answer to this question should also be empirically demonstrated.

#############################################################################################

Additional feedback
1. This benchmark could very well be a valuable contribution that fills a hole in the existing body of work, but the paper in its current form does not adequately establish this. The rebuttal should better address how this benchmark fits into existing work by comparing it to existing datasets and more relevant baselines.
2. The paper as a whole is well written, but I question some of the choices in syntax: terms such as “demarcation” and “desideratum” are spirited but may be better replaced by plainer alternatives.

---

> ### Author Response · Authors · 2020-11-24
> **authors' response**
>
> We wholeheartedly thank the reviewer for such a thoughtful review. It was refreshing to read the positive aspects of the paper laid out in a way that legitimately reflect the reviewer's own appreciation of the work. Furthermore, the negative aspects have served us to refine our arguments on the revised submission (mostly, the abstract and introduction), for which we are also thankful. To address these latter remarks:
>
> 1. We are not sure of whether this means that the construction of the dataset is incremental, or even the whole notion of tackling the construction of this dataset is incremental. While the former could be true, the fact that Online Continual Learning continues to be focused on settings lacking all the desired features that we discuss shows that it is not incremental with respect to the state of the art, as it argues for a very different approach from the one that is prevalent when evaluating these systems.
> 2. True. We do not fully grasp however why this is a reason for rejection.
> 3. The issue here is not that the current models fail. The problem is that for most cases they cannot even be applied to a data stream with temporal dependencies. The reason is that the standard setting is one on which each example is self-contained and independent of the previous history. However, for language modelling, capturing temporal dependencies is crucial, but no model that we are aware of even has a notion of context that would allow us to apply it in our setting. Consider applying a model like GEM, which relies on storing memories. How should we decide to chunk the data stream? What shall we use as the hidden vector? All these decisions are non-trivial and research questions on their own.
>
> Regarding the questions, we cannot apply Masson d'Autume model for the same reasons of point 3 above. Namely, that model was conceived for sentence-classification / QA datasets, where each training instance is independent from each other. This should probably also address your second question.
>
> Regarding the additional feedback, we hope the revised version does a better job at establishing the hole that this work is filling. We also take note of your vocabulary suggestion, and while we have replaced "demarcation" for the simpler "delimitation", we could not find a good alternative for "desideratum" that expresses the same concept.

---

### Official Review · AnonReviewer4 · 2020-10-29
**The paper designs a benchmark CALM for evaluating online continual learning. It adds a third dimension, temporally situated evaluation, to the existing evaluation benchmark. In addition, new metrics, Loss after switch and Recovery time after switch, are proposed to study catastrophic forgetting. Finally, evaluate multiple baseline models based on the composition of experts.**

**Rating:** 4
**Confidence:** 4

**Review:**

Strengths:
This paper proposes a new evaluation framework and gives two available evaluation datasets
Weakness：
- the paper needs a major rewrite to improve fluency and to better state motivation and contribution
- the empirical validation is weak.

Reasons for accept:
The advantages of this paper are:
1)	this paper proposed a new evaluation benchmark and dataset to promote the related research of online continual learning;
2)	the proposed plastic gate allows it to distribute different distributions among different experts, which has certain effects from the experimental results.

Reasons for reject:
The shortcomings of this paper are:
1.	This paper is not enough novel and has not contributed enough to continual learning related research;
2.	The core motivation of this paper is not clear enough. The abstract mentioned that "it is hard to demarcate task boundaries in actual tasks", and then said that a new benchmark, new metrics, and gating technique are proposed. Stacked statements like this can hardly capture the main problem to be solved.
3.	The advantages of the new metrics are not clear. Because from the experimental results, PPL and PPL@sw have a strong correlation. Therefore, please explain its advantages in detail (including the advantages of this evaluation framework compared with the evaluation framework of related literature, and verify it)
4.	The baseline uses LSTM and does not use CNN, Transformer, .etc, which shows that its generalization is limited.
5.	Can you provide the experimental results when λ is other values, and the combination of the number of modules？
6.	Because what you are proposing is a continuous language modeling evaluation framework. Is it possible to evaluate some of the latest online continual learning systems?
For example：
1) Lifelong Machine Learning with Deep Streaming Linear Discriminant Analysis
2) Learning a Unified Classifier Incrementally via Rebalancing
Or other Task-Free Continual Learning related work. This will have a good evaluation effect on measuring the versatility of your evaluation framework.

---

> ### Author Response · Authors · 2020-11-24
> **rebuttal**
>
>
> We have revised the submission to streamline the core motivations of this work (mainly, abstract and introduction). In what follows, we discuss the reviewer's reasons to reject:
>
> 1. It is hard to argue with this statement. What is enough?
> 2. The core problem is the scarcity of benchmarks for Online Continual Learning on a domain with temporally-dependent data, such as the linguistic domain. We have clarified this in the abstract and the second paragraph of the introduction.
> 3. We have revised the submission to clarify that the metrics are intended as a complement of the standard online performance, and not to substitute alternatives. Their goal is simply to zoom in into the behaviour at the switching points, which we discuss extensively.
> 4. The model is fully general to support any kind of module. We did conduct experiments with LSTMs because, as we reported, they worked much better out-of-the-box than Transformers in this setting. We did not conduct experiments on CNNs, but then again, nothing prevents the gating mechanism being applied to these kinds of experts.
> 5. We are not sure to understand what the reviewer means here. We experimented with two settings for the $\lambda$ parameter on the multilingual and multidomain datasets, and for each of these, two levels of aggregation on the number of modules.
> 6. Thanks for the references. As it happens with other methods, we cannot directly port these methods to our setting because they all assume that each instance is presented as a self-contained history-independent example $x_t$, whereas learning to represent the history is a fundamental aspect of learning in a linguistic setting. We believe that our contribution will allow the continual learning community to devote some attention to this problem which is obscured by the current most commonly used benchmarks.

---

### Official Review · AnonReviewer3 · 2020-10-30
**Hard to follow, imprecise statements, no comparison with old continual learning approaches**

**Rating:** 3
**Confidence:** 4

**Review:**

Summary: The paper proposes two benchmarks for continual language modeling: one evaluating character-level multilingual drift between languages which share similar characters and second evaluating word-level drift between English corpora of different domains. The setup is online in the sense of evaluation: they evaluate on the new sentences and then train over them (unlike image datasets), and catastrophic forgetting is hence characterised as having higher error than was in the past when there is a switch between the domains/languages. Hence, the loss functions measuring forgetting quantify the height and length of the rise in error. They compare a mixture of expert baselines with gating by different gating methods on this setup.


Primary Concerns:

1. There are few  sentences and terms that are hard to understand and to me they seem imprecise. Examples would be:

    (1.1) Intro: “human children still manage to acquire multiple languages without being explicitly asked to keep them separated” -- not sure if I buy this as it is known that if children are exposed to situation where there are many languages, they get confused, sometimes many kids find it hard to learn any of them, and it becomes important to give them guiding signal. Do you have any reference to support this hypothesis?

    (1.2) Section 3, second para: “preventing data leakage”: what do you mean by data leakage?

    (1.3) Section 3, third para: hard to follow, notation isn’t clear. And it seems there is a typo in S_i = \sum_j T_i.

    (1.4) Section 3, fourth para: “for a model to be resilient to forgetting, it must adapt quickly”: this statement is not correct because if a model adapts quickly to a new distribution, the parameter change would lead to forgetting and that’s primary the reason why there are regularization based approaches for continual learning enforcing models to be in the vicinity of old parameters. Too much adaptivity does not ensure less forgetting.

    (1.5) Section 3, loss after switch: what do you mean by a switch? How do you know when a switch happens (task label is not given)? In practice the loss curve is not smooth. How do you identify the switch? Fig 1 (a) is too smooth, does not represent the real loss curve.

2. Regarding experiments, is it not possible to design much simpler methods which work for this problem? If it's known there is expected to be a character/word-sequence distribution shift, I believe it's likely they can be detected easily with traditional n-gram models and style distinguishing attributes typically used for author identification [1,2]. Why isn't it possible to use a baseline which consists of experts for one domain/language where the character-sequence decides which expert to use instead of these weaker gating-based methods? Also, English/czech/german/french seem very distinguishable and share little in common in terms of character sequences [3], hence I am doubtful of the finding that combining these models will improve any single language performance.

3. Why is it not possible to apply traditional continual methods like Experience replay to this setting-- you simply store intelligently selected past sentences in memory (when say error shoots up) and replay using them. There are many other continual learning approaches that potentially could be applied here. Any particular reason for not using them?

[1] Koppel et. al., Computational Methods in Authorship Attribution
[2] Sapkota et. al., Not All Character N-grams Are Created Equal: A Study in Authorship Attribution
[3] Gerz et. al., On the Relation between Linguistic Typology and (Limitations of) Multilingual Language Modeling (edited)

---

> ### Author Response · Authors · 2020-11-24
> **rebuttal**
>
> We are sorry to hear the reviewer found the paper hard to follow, but please note that R1 and R2 have exact opposite views on this.
> 1. As for what the reviewer considers imprecise statements:
>
>     (1.1) The same reference that we cite in the paper begins Section 2 points out that bilingualism is a widespread phenomenon, which is the norm rather than the exception among children. What we meant here was that even if they rely on any cue, this cue must necessarily be learned. We have further clarified this in the submission.
>
>     (1.2.) Data leakage is a fairly standard term referring to a situation where you train on test data.
>
>     (1.3.) Thanks for pointing this out, we have improved the presentation of this paragraph in the revised submission.
>
>     (1.4.) We understand that this might sound counterintuitive given the usual framing in the literature, but consider that the best way to adapt quickly to a distribution that you have actually seen in the past is to _remember_ about it, rather than re-learn it. Perhaps that in this light, the stability-plasticity dilemma could look less of a dilemma.
>
>     (1.5.) We know when switches happen as evaluators, but this information is hidden from the models, analogously to how you hide test labels when evaluating on a regular classification setting.
> 2. One could try to apply the techniques that the reviewer is proposing for the multilingual setting, but then how would they generalize to the multidomain? It is important not to miss from sight the long-term goal of developing general learners on multiple types of data. Here we are contributing a new benchmark on a linguistic setting, but it is not an end-goal per se. As for the one-expert-per-class suggested baseline, we do have an even stronger baseline, which is given by the independent LSTM. In contrast to what the reviewer is suggesting, we do not claim that the gating model benefits from the multilingual setting, in which it doesn't, but rather that it displays lower levels of forgetting than other alternatives. In contrast, the gating model does benefit from transfer on the multidomain setting. See last paragraph of Section 5.1 for a discussion that addresses this point.
> 3. Because it is far from obvious how that should be done in language, and it is a research question that goes beyond the scope of this single paper. While an instance or two of how to draw the number 5 can be a good representation of a class, which sentences can be a good representation for a full language? Moreover, how should you chunk the input into an experience? Note that one of the crucial points that distinguish language from the image domain is that while the latter is naturally chunked into independent examples, language occurs as a continuous stream of correlated data.

---

### Decision · Program_Chairs · 2021-01-07
**Final Decision**

**Decision:**

Reject

**Comment:**

The initial reviews were mixed for this paper. On one hand, some of the reviewers highlighted that the proposed datasets could be useful to researchers. On the other, reviewers found a few important flaws with the current manuscript including missing baselines, issues with the proposed tasks, and possibly inaccurate/imprecise statements.

Our discussion after the author's response focussed on whether the positives aspects of the current paper outweighed some of the perceived weaknesses of the paper. In particular, while some of the initial criticisms from the reviewers were successfully addressed by the authors (including possible imprecisions and to a certain extent motivation), all the reviewers remained convinced that standard continual learning baselines could be adapted to this setting. They also conjectured that these missing baselines might not allow readers to appreciate the strength of the proposed datasets.

In their response, the authors argued that adapting models would require research. The reviewers are under the impression that it would be useful to test baselines more or less "as-is" even if the authors do not think these baselines will be competitive. For example, in the discussion, a reviewer suggested that "an experience replay baseline could [...] have been implemented" where the replay buffer includes the hidden states of an LSTM. It might also be useful to study baselines that do not strictly obey the proposed setting, again to get a better understanding of the proposed tasks (including how difficult it is).  Overall, having some of these baselines would be one way to better connect the proposed work to the current continual-learning literature.